# Genetic Control of Photosynthesis in Sugarcane During Successive Ratoon Cycles

**DOI:** 10.3390/biology15010075

**Published:** 2025-12-31

**Authors:** Chi Zhang, Yibin Wei, Yuzhi Xu, Abdullah Khan, Chunxiu Jiang, Huojian Li, Jun Chen, Yuling Wu, Zuli Yang, Jiafu Chen, Fangmei Liang, Jianlong Xu, Muqing Zhang, Yixue Bao

**Affiliations:** 1College of Agriculture, Guangxi Key Laboratory of Sugarcane Biology, Guangxi University, Nanning 530005, China; 18740375350@163.com (C.Z.); 2117301048@st.gxu.edu.cn (Y.W.); xuyuzhi1992@163.com (Y.X.); abdullah.pbg@aup.edu.pk (A.K.); jchunxiu99@163.com (C.J.); lhj1430304146@163.com (H.L.); cj265467@163.com (J.C.); 15777188915@163.com (Y.W.); 2Laibin Academy of Agricultural Sciences, Laibin Branch of Guangxi Academy of Agricultural Sciences, Laibin 546100, China; yang_zuli@163.com (Z.Y.); chenhb123c@163.com (J.C.); 3Liucheng County Sugarcane Research Center, Liuzhou 545200, China; liangfangmei202105@163.com; 4Institute of Crop Sciences, Chinese Academy of Agricultural Sciences, Beijing 100081, China; xujianlong@caas.cn; 5Hainan Seed Industry Laboratory, Sanya 572024, China

**Keywords:** sugarcane, photosynthetic traits, photosystem II, sugarcane quality

## Abstract

Sugarcane (*Saccharum* spp.) is a globally significant crop for both sugar production and bioenergy generation. Investigations into its photosynthetic characteristics and underlying genetic mechanisms are of paramount importance for yield enhancement and quality improvement. This study demonstrates that genotype constitutes a critical determinant in modulating sugarcane’s photosynthetic traits. The canal point series cultivars introduced from the United States demonstrate superior photosynthetic efficiency, thereby presenting substantial value for breeding programs aimed at developing sugarcane varieties with enhanced photosynthetic performance. By systematically dissecting these photosynthetic determinants, this work establishes a robust theoretical foundation for developing next-generation high-photosynthesis cultivars, thereby paving the way for optimized sugarcane cultivation systems worldwide.

## 1. Introduction

Global agricultural productivity currently confronts multifaceted challenges [1,2]. As a strategically important crop, sugarcane serves dual roles as both a staple sugar source and a renewable biofuel feedstock [3]. Chinese sugarcane breeders have successfully developed cultivars adapted to diverse ecological regions through interspecific hybridization, utilizing native wild relatives (*Saccharum spontaneum* L. and *Erianthus arundinaceus* (Retz.) Jeswiet) alongside introduced germplasm resources. The United States maintains the world’s most comprehensive sugarcane germplasm repository. The Canal Point (CP) breeding program has produced elite parental lines renowned for high yield, superior sucrose content, strong ratooning ability, and enhanced stress resistance. Over decades of breeding practice in China, CP-derived cultivars have been extensively utilized as parental materials, contributing significantly to the development of numerous improved sugarcane varieties. These advances have played a pivotal role in strengthening China’s sucrose industry and ensuring its sustainable development. Research on sugarcane photosynthetic traits and their genetic mechanisms has consequently gained critical importance for optimizing production efficiency [4,5,6]. Given that sugarcane yield and quality directly influence global sugar security and energy sustainability, elucidating the physiological processes—particularly photosynthetic mechanisms—is imperative for crop improvement. This knowledge further supports the transition towards sustainable agricultural development worldwide. Photosynthesis represents the cornerstone of plant growth and development, governing light energy utilization and conversion. Newly planted sugarcane refers to the planting pattern of sugarcane established by direct sowing of sugarcane seed pieces (seedcane). Sugarcane ratooning cultivation refers to an agronomic practice where the underground root system is retained after the previous season’s harvest, utilizing the surviving buds to regenerate a new crop cycle. This method offers significant advantages, including reduced production costs, extended effective growth periods, enhanced early-season sugar accumulation, and improved soil structure conservation. Investigating sugarcane photosynthetic traits under plant and ratoon cane conditions—including photosynthetic rates and chlorophyll dynamics—provides essential scientific foundations for enhancing photosynthetic efficiency. Chlorophyll fluorescence parameters, as non-invasive probes of plant physiological status [7], have been extensively applied across diverse crop research domains [8,9,10]. Photosynthetic activity is strongly influenced by a range of environmental factors, including core determinants such as light intensity, CO_2_ concentration, and temperature, along with key regulatory elements like water availability and mineral nutrients. Hegde et al. [11] demonstrated that leaf water deficit in sugarcane leads to significant reductions in both variable fluorescence (Fv) and the maximum photochemical efficiency of PSII (Fv/Fm). Qi et al. [12] reported a significant reduction in leaf chlorophyll content in rice under low nitrogen supply. Key parameters include: Fo (Initial fluorescence), Fm (Maximum fluorescence), Fv (Variable fluorescence), Fv/Fm (Maximum quantum yield), and Y(NO) (Non-regulated energy dissipation quantum yield) [13,14].

‌Fo represents the basal fluorescence intensity when photosystem II (PSII) reaction centers are fully open. It reflects the electron density excited by PSII antenna pigments and is positively correlated with chlorophyll concentration. Genotypes with higher photosynthetic efficiency typically demonstrate lower Fo values, suggesting more effective light energy capture by chlorophyll antenna systems. Fm denotes the maximum fluorescence intensity when PSII reaction centers are fully closed, serving as an indicator of PSII electron transfer capacity. High-efficiency genotypes characteristically exhibit elevated Fm values, implying superior electron transfer efficiency within PSII [15]. Fv represents the maximum variable fluorescence intensity in dark-adapted leaves, reflecting the reduction state of PSII primary electron acceptors. Genotypes with enhanced photosynthetic efficiency generally display increased Fv values, indicating greater acceptor availability and improved light energy utilization. The ratio Fv/Fm quantifies the maximum quantum yield of PSII, representing the plant’s theoretical photosynthetic potential. Higher Fv/Fm values in high-efficiency genotypes signify greater photosynthetic capacity. Similarly, Fv/Fo serves as an indicator of PSII electron transfer efficiency and potential activity. Elevated Fv/Fo values in high-efficiency genotypes suggest enhanced electron transport through PSII. Y(NO) measures light-induced damage, with higher values indicating incomplete photochemical energy conversion and insufficient protective mechanisms to dissipate excess energy. High-efficiency genotypes typically show reduced Y(NO) values, demonstrating better light energy utilization and decreased energy wastage [16]. Simultaneously, investigating the genetic mechanisms of sugarcane photosynthetic traits through the analysis of key gene expression and regulation can provide precise genetic insights for breeding programs. This approach will accelerate the development of more adaptable and high-yielding sugarcane varieties [17,18,19].

This study aims to elucidate the intricate relationships between photosynthetic characteristics and their underlying genetic mechanisms in sugarcane. By comprehensively analyzing genotype–ratoon age interactions, our research seeks to establish a robust scientific foundation for improving sugarcane productivity and promoting sustainable industry development.

## 2. Material and Methods

### 2.1. Tested Sugarcane Genotypes and Field Experimental Design

This study evaluated a total of 74 sugarcane genotypes, including 20 genotypes from the United States, India, and Australia, along with 54 genotypes from seven provinces in China: Guangxi (26), Fujian (10), Guangdong (6), Yunnan (4), Taiwan (5), Hainan (2), and Sichuan (1) (Appendix A). All materials were collected and maintained by the Sugarcane Germplasm Innovation and Genetic Improvement Research Team at Guangxi University, and are preserved in the sugarcane germplasm repository. Field trials employed a completely randomized block design with single-row plots (5 m length, 1.4 m row spacing) at 16 shoots per meter. All genotypes were replicated three times. The experiment was conducted at Guangxi University’s Fusui Field Station, which is located at 107°48′ E, 22°34′ N, with an altitude of 69.5 m, the field site has an average annual sunshine duration of 1693 h, a mean frost-free period of 346 days, and total solar radiation of 108.4 kcal/cm^2^. The field soil is classified as red soil, with sugarcane as the previous crop. Commercial compound fertilizer with a total nutrient content of 42% (N-P-K) was adopted for fertilization, which was applied in a single application at the tillering stage for both newly planted and ratoon sugarcane; the application rate was 1200 kg per hectare for newly planted sugarcane and 1000 kg per hectare for ratoon sugarcane. The field experiment was initiated by planting plant cane in February 2021; the aboveground parts were harvested in January 2022, with the first ratoon crop (R1) retained. After harvesting the aboveground parts of R1 in January 2023, the second ratoon crop (R2) was maintained, and all data related to R2 were collected within 2023.

### 2.2. Field Data Collection

The Fo, Fm, and SPAD values were measured during the sugarcane elongation stage (six months after planting/harvest) under consecutive sunny conditions. The youngest, newly emerged and unexpanded leaf at the apex is referred to as the spindle leaf (heart leaf). The first fully expanded leaf immediately below the spindle is designated the +1 leaf, the second one below the +1 leaf the +2 leaf, and the third one below the +2 leaf the +3 leaf. The +1, +2, and +3 leaves are the core functional leaves of sugarcane and accurately reflect the photosynthetic capacity of the plant. The chlorophyll fluorescence parameters (Fo, Fm) of the +1, +2, and +3 leaves were measured between 20:00 and 23:00 at night using a portable pulse-amplitude modulation chlorophyll fluorometer, PAM-2500 (Heinz Walz GmbH, Effeltrich, Germany) [9]. During the daytime, the relative chlorophyll content (greenness index) of the same leaves (the +1, +2, and +3 leaves) was determined using a chlorophyll meter, SPAD-502 Plus (Konica Minolta, Tokyo, Japan). For each sugarcane genotype, 10 healthy plants were selected; Fo, Fm, and SPAD values were measured on the +1, +2, and +3 leaves of each plant, with three technical replicates set for each leaf.

### 2.3. Sugarcane Quality Assessment

In December 2021 and December 2022, during the sugarcane maturation period, six representative stalk samples were selected from each tested material for subsequent analysis. The fresh weight of sugarcane stalks was measured using an electronic balance, and the single-stalk weight of each genotype was calculated. Sugarcane stalks were crushed using the DM540-CPS Sugarcane Crushing System (Sugarcane Crushing System, IRBI, Araçatuba, Brazil), and sucrose content was determined using the MATRIX-F system (Bruker Optik GmbH, Ettlingen, Germany) [20].

### 2.4. Statistical Analysis

Phenotypic data were processed in Excel 2021 (Microsoft, Redmond, WA, USA) for preliminary organization. Mixed-model variance analysis was implemented via the “lme4” package (Version 1.1-35) in R 4.3.1 (R Foundation for Statistical Computing, Vienna, Austria) to account for genotype-year interactions. Principal component analysis (PCA) of photosynthetic traits was conducted using “FactoMineR” package (Version 2.4) in R 4.3.1, incorporating both annual means and multi-year aggregates. Subsequent clustering and discriminant analyses were performed in SPSS 2022 (IBM, Armonk, NY, USA) based on principal component scores. Post hoc multiple comparisons employed Fisher’s least significant difference (LSD) test (α = 0.05). Visualization was achieved using Origin 2022 (OriginLab, Northampton, MA, USA).

## 3. Results

### 3.1. Joint Variance Analysis of Sugarcane Photosynthetic Traits

Comprehensive analysis of key photosynthetic indicators, including chlorophyll fluorescence parameters (Fv/Fo, Fv/Fm, Y(NO)) and relative chlorophyll content (SPAD values), was conducted across the evaluated sugarcane germplasms (NP, R1, R2). The data demonstrated normal distribution patterns (Shapiro–Wilk test, *p* > 0.05) for all measured traits, with newly planted materials (NP) exhibiting significantly higher photosynthetic efficiency than ratooned groups (SPAD: 44.72 ± 4.68 vs. 42.64 ± 4.79) (Figure 1). Joint variance analysis using mixed-model ANOVA revealed highly significant effects (*p* < 0.001) for both genotype (F = 73) and ratoon year (F = 2) factors, accounting for 37.43% and 0.68% of total variance, respectively. Crucially, the genotype × ratoon year interaction term was also significant (F = 146, *p* < 0.01), explaining 22.19% of variance, indicating that repeat factors differentially influence photosynthetic performance among genotypes. Measurements across three growing cycles showed minimal variation (coefficient of variation < 0.05%) for all traits, validating the experimental design’s reliability (Table 1). Furthermore, the heritability estimates for all photosynthetic parameters exceeded 70%.

### 3.2. Principal Component Analysis of Sugarcane Photosynthetic Traits

Principal component analysis (PCA) was performed on standardized photosynthetic trait data from 74 sugarcane genotypes, including newly planted (NP) and ratooned for first-year ratoon (R1) and second-year ratoon (R2) crops over three consecutive growing seasons (Figure 2). The first three principal components (PC1–PC3) accounted for 99.9% of the total phenotypic variance (Kaiser criterion), effectively compressing the original measured parameters into interpretable composite indices (Appendix A). PC1 (eigenvalue = 3.218, explaining 46.03% of variance) was strongly correlated with photosynthetic efficiency parameters (Fv/Fm, r = 0.987; Fv/Fo, r = 0.984) and dark-adapted quantum yield (Y(NO), r = −0.987), likely representing the overall photosynthetic capacity and light energy utilization efficiency. PC2 (eigenvalue = 2.866, explaining 41.00% of variance) was dominated by fluorescence parameters Fo (r = 0.977), Fm (r = 0.987), and Fv (r = 0.947), suggesting its association with reaction center integrity and electron transfer chain functionality. PC3 (eigenvalue = 0.907, explaining 12.98% of variance) exhibited high loadings on SPAD values (r = 0.933), serving as a proxy for chlorophyll content dynamics. This dimensionality reduction approach not only confirmed the multicollinearity among photosynthetic traits but also revealed that photosynthetic efficiency (PC1) constitutes the primary adaptive trait, electron transport stability (PC2) shows genotype-specific responses to ratooning, and chlorophyll metabolism (PC3) exhibits distinct year-to-year variation patterns.

### 3.3. Cluster Analysis and Discriminant Analysis of Sugarcane Photosynthetic Traits

Cluster analysis was performed using Ward’s method with Euclidean distance as the similarity measure. The 74 sugarcane genotypes were classified into three categories: High Photosynthetic Efficiency (HPE), Moderate Photosynthetic Efficiency (MPE), and Low Photosynthetic Efficiency (LPE), accounting for 60.81%, 25.68%, and 13.51% of the total genotypes, respectively (Figure 3; Table 2). Further multiple comparison analysis revealed significant differences in photosynthetic trait indicators among these categories (Table 3), which facilitates the analysis of photosynthetic parameter variations among different genotypes.

High photosynthetic efficiency genotypes exhibited excellent photosynthetic performance, primarily characterized by efficient photosystem II (PSII) photochemical efficiency, higher Fv/Fm and SPAD values, and lower Y(NO) values. In contrast, low photosynthetic efficiency genotypes showed poor performance in these parameters, possibly due to deficiencies in PSII leading to decreased light energy conversion efficiency. Moderate photosynthetic efficiency genotypes exhibited intermediate performance, showing moderate initial fluorescence, maximum fluorescence, and PSII maximum quantum yield, a relatively balanced electron transfer situation, and moderate light energy conversion efficiency and quenching mechanisms.

### 3.4. Quality Analysis of Genotypes for Photosynthetic Efficiency in Sugarcane

In this study, sugarcane genotypes were systematically selected from three distinct photosynthetic efficiency categories: HPE (YZ99-596, FR99-49, GT02-761, GT05-322, FN04-1027), MPE (GUC41, GT08-297, CP96-1602, MT11-610, YZ89-159), and LPE (CP011372K, GT02-467, FN40, GT05-3846, YG24). The research primarily assessed their agronomic performance through two key quality parameters (single-stalk weight and sucrose content). Statistical analysis was performed using Tukey’s honest significant difference (HSD) test at the 0.05 significance level (Figure 4). The analysis revealed that genotypes exhibiting higher photosynthetic efficiency consistently outperformed their counterparts: Compared to the LPE group, the HPE group exhibited significantly greater single-stalk weight (2.11 ± 0.27 kg vs. 1.40 ± 0.24 kg) and sucrose content (16.55 ± 1.64% vs. 12.69 ± 0.89%). Correlation analysis was performed by integrating key photosynthetic parameters (Fm, Fo, Fv, Fv/Fm, Fv/Fo, Y(NO), SPAD) with agronomic traits including single-stalk weight (SSW) and sucrose content (SC). SSW showed significant positive correlations with Fo and Y(NO), but negative correlations with SPAD, Fv/Fm, and Fv/Fo. Similarly, SC demonstrated positive associations with Fo, Fm, Y(NO), and Fv, while exhibiting negative relationships with SPAD, Fv/Fm, and Fv/Fo (Figure 5). These results unequivocally demonstrate a strong correlation between photosynthetic capacity and sugarcane quality, highlighting the potential for improving yield and sucrose accumulation through targeted breeding strategies.

## 4. Discussion

Photosynthesis, the fundamental mechanism through which plants harness solar energy for primary production, establishes photosynthetic light-use efficiency as a critical determinant of crop yields. The current study assessed the photosynthetic parameters of various sugarcane genotypes over multiple years. The results revealed a significant impact of genotype on the photosynthetic performance of sugarcane. The discerned variations in photosynthetic traits among distinct genotypes highlight the diversity in sugarcane germplasm concerning photosynthesis [21]. Combined analysis of variance revealed significant differences in sugarcane photosynthetic traits among years and genotypes. Additionally, we found that ratoon age significantly affected the photosynthetic traits of sugarcane. Newly planted sugarcane exhibited superior performance in chlorophyll fluorescence parameters and relative chlorophyll content compared to ratoon cane. This difference may be attributed to the gradual aging of the plant’s photosynthetic mechanism with an increase in ratoon age [22]. Wei et al. [23] investigated photosynthetic characteristics across 258 sugarcane cultivars over three consecutive years at different leaf positions. The results demonstrated significant variations in photosynthetic traits among genotypes, years, and leaf positions. Chlorophyll fluorescence parameters (Fo, Fm, and Fv) were consistently higher in plant cane than in ratoon cane, which aligns with the findings of our current study, confirming that sugarcane photosynthesis is influenced by both genotype and year. These findings are practically significant for optimizing sugarcane planting management strategies.

Photosynthetic performance demonstrated close associations with quality traits in sugarcane. High-photosynthetic-efficiency genotypes exhibited greater single-stalk weight and sucrose content, supporting photosynthesis-driven breeding for quality improvement [24]. In a related study, Wang et al. [25] classified 60 rapeseed varieties into three groups based on light-use efficiency: six showed high efficiency, 18 moderate efficiency, and 36 low efficiency. The U.S. CP series used in this trial displayed superior photosynthetic characteristics under local conditions, highlighting their value as parental lines in hybridization programs. These findings further suggest that enhancing photosynthetic performance represents an effective strategy for optimizing sugarcane quality. Future studies should focus on evaluating the adaptability and stress tolerance of such germplasm across diverse environments to improve trait precision and global suitability in sugarcane breeding.

Sugarcane varieties cultivated in Hainan and Guangxi exhibited superior performance over three consecutive years, suggesting that geographical location significantly impacts sugarcane adaptability and productivity. Therefore, these regional differences must be thoroughly considered during the breeding process. Specific genotype combinations may exert synergistic effects to enhance photosynthetic efficiency, whereas unfavorable combinations may induce negative interactions that reduce photosynthetic performance [26]. Further molecular genetic studies are needed to elucidate the specific interactions between genotypes [27]. Understanding the genetic mechanisms of sugarcane photosynthetic traits is crucial for future breeding efforts. Molecular marker-assisted breeding techniques can help identify key genes associated with photosynthetic traits, enabling the selection of genotypes more favorable for high photosynthetic efficiency [28]. This precision breeding approach could expedite the development of sugarcane varieties with higher yields and greater adaptability [29].

When delving into the genetic mechanisms of sugarcane photosynthetic traits, it is crucial to consider the impact of environmental factors on genetic expression [26]. Certain genotypes may exhibit better photosynthetic performance under specific environmental conditions while performing poorly under other conditions [30]. Therefore, future research should also consider the regulatory effects of environmental factors such as light, temperature, and moisture on photosynthetic traits and focus on identifying specific genetic variation loci, exploring networks of interactions between genes, and incorporating research in epigenetics, among other aspects, to unveil more profound layers of regulation in sugarcane photosynthetic traits [31]. This holds the potential to provide more precise guidance for future improvements and optimization in sugarcane breeding [32].

## 5. Conclusions

In conclusion, our comprehensive investigation into sugarcane photosynthetic traits provides critical insights for both cultivation management and genetic enhancement strategies, systematically elucidating the complex tripartite relationship between genotype, ratoon age, and germplasm origin in determining photosynthetic efficiency. Comprehensive analysis of 74 sugarcane genotypes revealed significant photosynthetic efficiency variation, with pronounced effects of ratoon age. Principal component analysis demonstrated the critical role of chlorophyll fluorescence parameters (Fo, Fv/Fo, Fv/Fm, Y(NO)) in explaining the variance in photosynthetic traits. Cluster and discriminant analyses classified the genotypes into high (60.81%), moderate (25.68%), and low (13.51%) photosynthetic efficiency groups. Notably, high photosynthetic efficiency genotypes exhibited strong correlations with enhanced single-stalk weight and sucrose content, providing valuable insights for improving sugarcane yield and quality through targeted breeding strategies.

## Figures and Tables

**Figure 1 biology-15-00075-f001:**
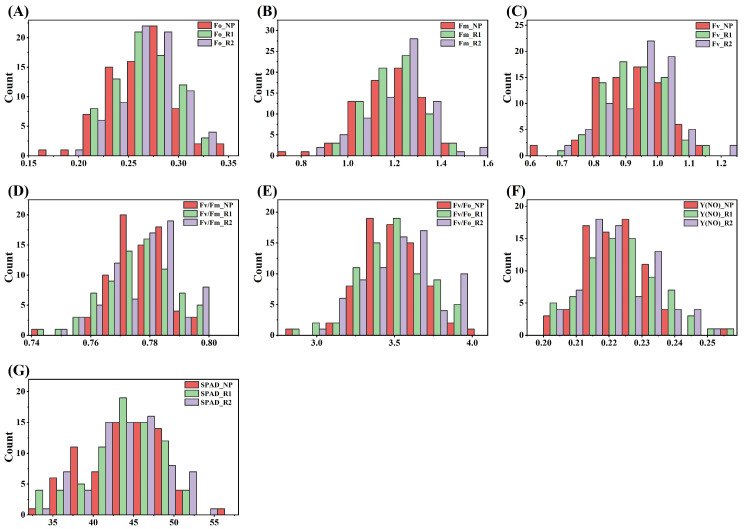
Comparative analysis of photosynthetic traits among NP, R1 and R2 sugarcane. (**A**–**G**) indicates the difference and distribution of photosynthetic characteristics of sugarcane in different years: Fo, Fm, Fv, Fv/Fm, Fv/Fo, Y(NO), SPAD, respectively. All parameters are dimensionless, expressed as ratios or relative values.

**Figure 2 biology-15-00075-f002:**
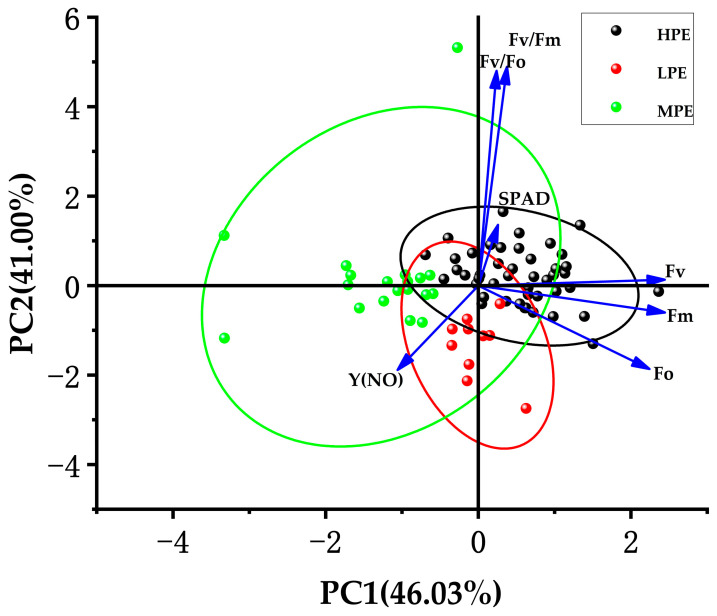
Principal component analysis for photosynthetic efficiency and factor weight of sugarcane. Arrows represent the loadings of each photosynthetic trait on PC1 and PC2; ellipses indicate the sample clustering ranges of different photosynthetic efficiency groups (HPE: high photosynthetic efficiency group; LPE: low photosynthetic efficiency group; MPE: medium photosynthetic efficiency group).

**Figure 3 biology-15-00075-f003:**
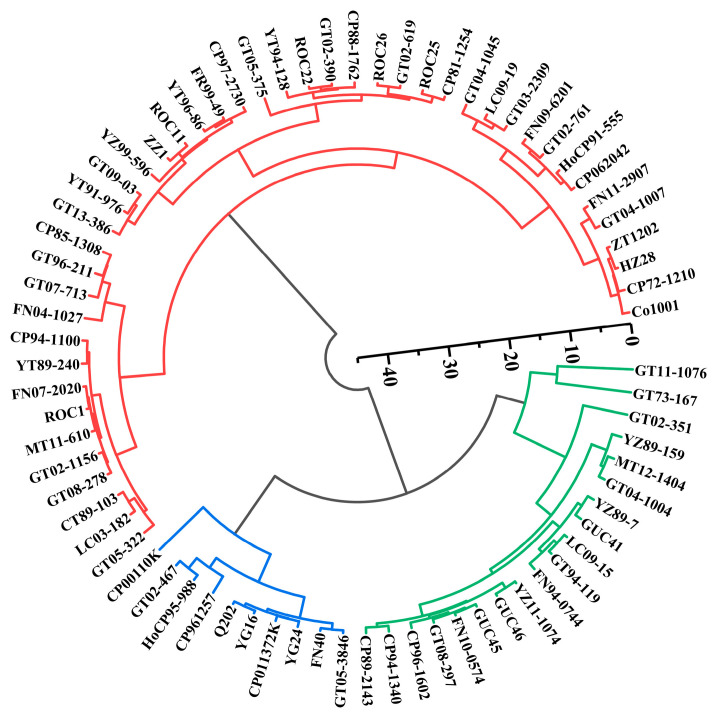
Dendrogram-based clustering of photosynthetic performance in 74 sugarcane genotypes. Color-coded cluster groups represent photosynthetic capacity levels: red represents 45 germplasms with high photosynthetic characteristics, green represents 19 germplasms with medium photosynthetic characteristics, and blue represents 10 with low photosynthetic characteristics.

**Figure 4 biology-15-00075-f004:**
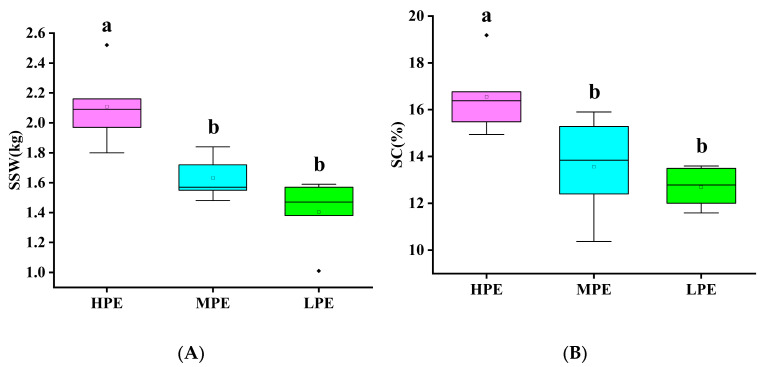
Yield-quality correlations stratified by photosynthetic efficiency in sugarcane germplasm. (**A**) Boxplot of single-stalk weight (kg) across photosynthetic efficiency groups; (**B**) Sucrose content (%) with significant group differences; ANOVA, *p* < 0.05; High Photosynthetic Efficiency (HPE), Moderate Photosynthetic Efficiency (MPE), Low Photosynthetic Efficiency (LPE); single-stalk weight (SSW), Sucrose content (SC); Lowercase letters a, b indicate significant differences at *p* < 0.05; Dots represent outliers in each group.

**Figure 5 biology-15-00075-f005:**
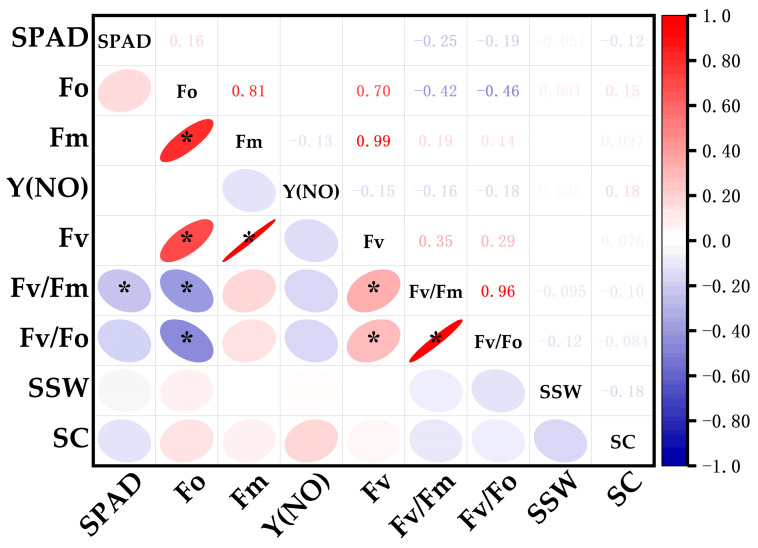
Correlation between photosynthetic parameters and single-stalk weight/sucrose content in tested materials. Color intensity and ellipse shape indicate the strength of the correlation (the darker the color or the flatter the ellipse, the stronger the correlation), * *p* ≤ 0.05.

**Table 1 biology-15-00075-t001:** Combined variance analysis for chlorophyll fluorescence parameters and relative chlorophyll content.

Source of Variation	Df	SPAD		Fo		Fm		Fv		Fv/Fm		Fv/Fo		Y(NO)	
		Sum Sq	SS%	Sum Sq	SS%	Sum Sq	SS%	Sum Sq	SS%	Sum Sq	SS%	Sum Sq	SS%	Sum Sq	SS%
Line	73	172.3 ***	61.50	0.006851 ***	46.87	0.11749 ***	43.88	0.07181 ***	40.74	0.0006805 ***	23.73	0.3618 ***	21.54	0.0006805 ***	23.73
Rep	2	11.7	0.11	0.000564	0.10	0.03095	0.32	0.02345	0.37	0.0002672	0.25	0.1667	0.27	0.0002672	0.25
Year	2	411.3 ***	4.02	0.00139	0.26	0.00996	0.10	0.00391	0.06	0.0000822	0.08	0.0826	0.14	0.0000822	0.08
Line:Rep	146	18.2 ***	12.98	0.00158 ***	21.61	0.0281 ***	20.99	0.01878 ***	21.31	0.0003598 **	25.10	0.2372 ***	28.25	0.0003598 **	25.10
Residuals	442	9.9	21.39	0.000752	31.16	0.01535	34.71	0.01097	37.67	0.0002408	50.84	0.1382	49.81	0.0002408	50.84
h^2^_B_ (%)		86.00		81.82		81.39		80.57		72.27		70.30		72.27	

*** *p* ≤ 0.001; ** *p* ≤ 0.01; h^2^_B_: the broad-sense heritability.

**Table 2 biology-15-00075-t002:** Cluster and differential analysis of photosynthetic characteristics of sugarcane genotypes.

NO.	Genotype	SPAD	Y(NO)	Fo	Fm	Fv	Fv/Fm	Fv/Fo	Clustering
1	Co1001	40.43	0.22	0.27	1.22	0.95	0.78	3.48	HPE
2	CP062042	47.23	0.22	0.27	1.15	0.90	0.78	3.66	HPE
3	CP81-1254	45.57	0.21	0.28	1.36	1.08	0.79	3.85	HPE
4	CP85-1308	47.13	0.23	0.33	1.41	1.08	0.77	3.27	HPE
5	CP88-1762	46.43	0.21	0.30	1.35	1.05	0.79	3.74	HPE
6	CP94-1100	48.63	0.24	0.29	1.30	1.02	0.78	3.60	HPE
7	CP97-2730	46.27	0.23	0.27	1.26	0.99	0.79	3.77	HPE
8	CT89-103	49.70	0.25	0.27	1.22	0.95	0.78	3.58	HPE
9	FN04-1027	49.10	0.22	0.33	1.52	1.19	0.78	3.65	HPE
10	FN07-2020	46.80	0.23	0.29	1.30	1.00	0.77	3.44	HPE
11	FR99-49	43.60	0.21	0.27	1.27	1.00	0.79	3.67	HPE
12	GT02-390	44.70	0.21	0.29	1.35	1.06	0.79	3.71	HPE
13	GT02-619	49.23	0.21	0.29	1.35	1.06	0.79	3.71	HPE
14	GT02-761	48.37	0.22	0.25	1.18	0.93	0.79	3.72	HPE
15	GT05-322	56.77	0.22	0.29	1.29	1.00	0.78	3.57	HPE
16	GT05-375	49.53	0.15	0.30	1.34	1.05	0.80	4.00	HPE
17	GT09-03	44.07	0.20	0.27	1.28	1.01	0.80	4.02	HPE
18	GT13-386	47.30	0.20	0.26	1.23	0.99	0.80	4.14	HPE
19	GT96-211	50.07	0.23	0.31	1.33	1.03	0.77	3.39	HPE
20	HoCP91-555	46.07	0.21	0.26	1.19	0.93	0.78	3.61	HPE
21	LC03-182	49.33	0.23	0.28	1.22	0.94	0.77	3.48	HPE
22	ROC1	49.33	0.23	0.29	1.30	1.00	0.77	3.42	HPE
23	ROC11	45.03	0.23	0.29	1.29	1.03	0.79	3.87	HPE
24	ROC22	43.80	0.21	0.29	1.34	1.05	0.79	3.67	HPE
25	ROC25	47.97	0.20	0.27	1.34	1.07	0.80	3.90	HPE
26	ROC26	50.27	0.21	0.29	1.36	1.06	0.79	3.73	HPE
27	YT89-240	47.93	0.22	0.30	1.35	1.05	0.78	3.56	HPE
28	YT91-976	43.33	0.21	0.26	1.25	0.99	0.79	3.94	HPE
29	YT96-86	45.70	0.21	0.27	1.27	1.00	0.79	3.71	HPE
30	YZ99-596	51.47	0.23	0.27	1.23	0.96	0.79	3.86	HPE
31	ZZ1	46.60	0.22	0.27	1.27	1.01	0.80	3.89	HPE
32	CP72-1210	39.57	0.23	0.27	1.26	0.99	0.78	3.70	HPE
33	FN09-6201	49.50	0.23	0.24	1.12	0.88	0.79	3.71	HPE
34	GT02-1156	47.00	0.22	0.28	1.26	0.98	0.78	3.47	HPE
35	GT03-2309	42.93	0.20	0.24	1.16	0.92	0.80	3.90	HPE
36	GT04-1007	41.13	0.22	0.29	1.29	1.01	0.78	3.52	HPE
37	GT04-1045	36.47	0.22	0.25	1.17	0.92	0.78	3.74	HPE
38	GT07-713	42.83	0.22	0.31	1.41	1.09	0.78	3.47	HPE
39	GT08-278	46.30	0.22	0.29	1.30	1.02	0.78	3.54	HPE
40	HZ28	44.50	0.22	0.26	1.22	0.96	0.78	3.65	HPE
41	LC09-19	41.47	0.21	0.25	1.21	0.96	0.79	3.87	HPE
42	MT11-610	50.13	0.22	0.29	1.31	1.02	0.78	3.50	HPE
43	YT94-128	42.07	0.22	0.30	1.39	1.09	0.79	3.68	HPE
44	ZT1202	41.67	0.21	0.26	1.22	0.96	0.78	3.65	HPE
45	FN11-2907	37.47	0.22	0.29	1.32	1.03	0.78	3.51	HPE
46	CP89-2143	48.10	0.22	0.26	1.11	0.85	0.78	3.61	MPE
47	CP94-1340	43.93	0.22	0.26	1.06	0.81	0.78	3.63	MPE
48	GT04-1004	51.87	0.23	0.23	0.97	0.75	0.77	3.40	MPE
49	GT73-167	44.17	0.21	0.23	1.09	0.92	0.84	6.09	MPE
50	MT12-1404	48.20	0.22	0.21	0.98	0.76	0.78	3.56	MPE
51	CP96-1602	46.73	0.22	0.23	1.05	0.82	0.78	3.51	MPE
52	FN94-0744	44.97	0.23	0.23	1.01	0.77	0.77	3.33	MPE
53	GT08-297	46.37	0.22	0.24	1.11	0.87	0.78	3.59	MPE
54	GT11-1076	45.73	0.56	0.21	0.92	0.71	0.78	3.47	MPE
55	GT94-119	50.20	0.23	0.24	1.07	0.83	0.77	3.42	MPE
56	GUC41	48.03	0.24	0.26	1.09	0.84	0.77	3.26	MPE
57	GUC46	46.37	0.23	0.25	1.14	0.88	0.78	3.48	MPE
58	LC09-15	48.50	0.23	0.24	1.05	0.81	0.77	3.38	MPE
59	YZ11-1074	45.47	0.22	0.25	1.12	0.87	0.78	3.47	MPE
60	YZ89-159	41.77	0.22	0.22	0.99	0.78	0.78	3.57	MPE
61	YZ89-7	41.73	0.23	0.26	1.12	0.86	0.77	3.32	MPE
62	FN10-0574	44.60	0.22	0.24	1.09	0.85	0.78	3.54	MPE
63	GT02-351	43.07	0.21	0.14	0.79	0.64	0.79	3.72	MPE
64	GUC45	44.03	0.22	0.23	1.11	0.86	0.78	3.48	MPE
65	CP00110K	40.03	0.22	0.28	1.26	0.98	0.78	3.52	LPE
66	GT02-467	35.70	0.26	0.29	1.22	0.93	0.76	3.20	LPE
67	Q202	41.67	0.24	0.28	1.20	0.92	0.77	3.33	LPE
68	YG16	43.27	0.23	0.27	1.20	0.92	0.77	3.36	LPE
69	CP011372K	41.20	0.24	0.28	1.17	0.90	0.76	3.21	LPE
70	CP961257	42.87	0.26	0.32	1.34	0.99	0.74	2.86	LPE
71	FN40	43.23	0.23	0.28	1.24	0.96	0.77	3.26	LPE
72	GT05-3846	44.97	0.24	0.29	1.23	0.94	0.76	3.25	LPE
73	HoCP95-988	34.23	0.29	0.30	1.23	0.94	0.76	3.13	LPE
74	YG24	38.20	0.23	0.27	1.18	0.91	0.77	3.32	LPE

**Table 3 biology-15-00075-t003:** Differences in photosynthetic characteristics among different experimental genotypes.

Grade of Photosynthetic Efficiency	High Photosynthetic Efficiency (HPE)	Moderate Photosynthetic Efficiency (MPE)	Low Photosynthetic Efficiency (LPE)
**Tested genotypes**	Co1001, CP062042, CP81-1254, CP85-1308, CP88-1762, CP94-1100, CP97-2730, CT89-103, FN04-1027, FN07-2020, FR99-49, GT02-390, GT02-619, GT02-761, GT05-322, GT05-375, GT09-03, GT13-386, GT96-211, HoCP91-555, LC03-182, ROC1, ROC11, ROC22, ROC25, ROC26, YT89-240, YT91-976, YT96-86, YZ99-596, ZZ1, CP72-1210, FN09-6201, GT02-1156, GT03-2309, GT04-1007, GT04-1045, GT07-713, GT08-278, HZ28, LC09-19, MT11-610, YT94-128, ZT1202, FN11-2907	CP89-2143, CP94-1340, GT04-1004, GT73-167, MT12-1404, CP96-1602, FN94-0744, GT08-297, GT11-1076, GT94-119, GUC41, GUC46, LC09-15, YZ11-1074, YZ89-159, YZ89-7, FN10-0574, GT02-351, GUC45	CP00110K, GT02-467, Q202, YG16, CP011372K, CP961257, FN40, GT05-3846, HoCP95-988, YG24
**Fm**	1.28 ± 0.08 ^A^	1.05 ± 0.09 ^B^	1.23 ± 0.05 ^A^
**Fo**	0.28 ± 0.02 ^A^	0.23 ± 0.03 ^B^	0.29 ± 0.01 ^A^
**Fv**	1.01 ± 0.06 ^A^	0.81 ± 0.07 ^C^	0.94 ± 0.03 ^B^
**Fv/Fm**	0.79 ± 0.01 ^A^	0.78 ± 0.02 ^A^	0.76 ± 0.01 ^A^
**Fv/Fo**	3.68 ± 0.18 ^A^	3.62 ± 0.61 ^A^	3.25 ± 0.17 ^B^
**Y(NO)**	0.22 ± 0.01 A	0.24 ± 0.08 ^A^	0.24 ± 0.02 ^A^
**SPAD**	46.02 ± 3.89 ^A^	45.99 ± 2.71 ^A^	40.54 ± 3.50 ^B^
**Total**	45	19	10

Uppercase letters A, B, and C indicate significant differences at *p* < 0.05.

## Data Availability

All data generated or analyzed during this study have been included in this published article and its Appendix A. This study does not involve human or animal subjects, so ethical review is not required.

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
