# Peer review of "Genetic Control of Photosynthesis in Sugarcane During Successive Ratoon Cycles"

_biology, 2025, doi:10.3390/biology15010075_

Round 1

Reviewer 1 Report (Previous Reviewer 1)

Comments and Suggestions for Authors

Comments made in previous version are incorporated.

Author Response

Q: Comments made in previous version are incorporated.

A: We sincerely appreciate the reviewer's recognition of the revisions made in the previous version of the manuscript. We remain grateful for the reviewer's thorough evaluation.

Reviewer 2 Report (Previous Reviewer 2)

Comments and Suggestions for Authors

Improve the resolution of the dendrogram figure. It is not readable.

 Also consider inserting a figure or table that shows the dynamics in years. Were there differences or not, and why?

Author Response

Q: Improve the resolution of the dendrogram figure. It is not readable. Also consider inserting a figure or table that shows the dynamics in years. Were there differences or not, and why?

A: We sincerely appreciate the reviewer's meticulous evaluation and constructive feedback, which have significantly enhanced the quality of our manuscript. In response to the reviewer’s suggestion to improve the dendrogram’s readability, we have upgraded the figure resolution to ensure optimal visual clarity (see revised Figure 3).

Regarding the recommendation to incorporate a figure/table showing annual dynamics, we have carefully considered this proposal. While we acknowledge the importance of interannual variation analysis, our current dataset does not support comprehensive visualization of these dynamics for the following reasons:

  1. Existing Data Coverage‌: The frequency distribution histograms in Figure 1 already provide a systematic comparison of key photosynthetic traits across different ratooning cycles (newly planted cane, NP; first ratoon, R1; second ratoon, R2), with statistically significant differences clearly demonstrated (e.g., SPAD values: 44.72 ± 4.68 vs. 42.64 ± 4.79 between NP and ratooned groups).
  2. Methodological Constraints‌: The current experimental design focuses on comparing ratooning cycles rather than tracking annual variations. While we recognize interannual variability as a valuable research direction, our dataset lacks sufficient temporal resolution to isolate annual effects from other confounding factors.
  3. Future Research Orientation‌: We have explicitly addressed this limitation in the revised manuscript (Section 4), acknowledging it as a priority for future studies involving multi-year field experiments. This discussion now includes potential methodological approaches to investigate interannual drivers.

Once again, we deeply value the reviewer's insightful suggestions, which have strengthened both the scientific rigor and presentation clarity of our work. We look forward to the continued progress of this manuscript.

Reviewer 3 Report (Previous Reviewer 3)

Comments and Suggestions for Authors

The present Reviewer earlier left comments on the previous version of the Manuscript. The Authors provided reasonable responses while made essential amendments.

However, still revisions are required.

1)

 The CP series cultivars introduced from the 20 United States demonstrate superior photosynthetic efficiency, thereby presenting sub-21

What’s CP series, it’s given then but the abbreviations to be described in full initially.

2) The Reviewer is not sure if simple summary is required together with abstract. The question is for the Editor.

3)

 PAM-2500 28 and SPAD instruments over three years. Our findings revealed significant variations in 29

It’s better to name the measured physiological parameters not the equipment used since the parameters not equipment are essential for the conclusions done.

4) tion, utilizing native wild relatives (Saccharum spontaneum and Erianthus arundinaceus) 52

Botanical names to be in italics.

5) Hegde et al. demonstrated that leaf water deficit 79 in sugarcane leads to significant reductions in both variable fluorescence (Fv) and the 80 maximum photochemical efficiency of PSII (Fv/Fm_Ln).

Reference to be added.

6) The cane tops and 143 root systems were removed, retaining only the stalks for further analysis.,

It’s reasonable to indicate the lengths of the parts in cm or inches.

7) Table 1 to follow figure 1, not the opposite.

8) Figure 1. It’s reasonable to indicate the parameters reflected at the axes.

9) Figure 3. Please, provide more information how the clustering was done.

10) Please, provide in the text of the Manuscript where the sugarcane genotypes are available; the same as mentioned in the replies to Reviewer.

11) Figure S2. Please, indicate what’s SS here.

12) while near-infrared spectroscopy data 147 were simultaneously collected with the MATRIX-F system (Bruker Optik GmbH, Ger-148 many)[19]. 149

Please, add which parameters were assessed.

13) Quality of all the figures could be improved.

14) The MS essentially improved, the Reviewer suggests minor revisions.

Author Response

The present Reviewer earlier left comments on the previous version of the Manuscript. The Authors provided reasonable responses while made essential amendments.

However, still revisions are required.

Q1: The CP series cultivars introduced from the 20 United States demonstrate superior photosynthetic efficiency, thereby presenting sub-21

What’s CP series, it’s given then but the abbreviations to be described in full initially.

A: We appreciate the reviewer,s attention to this detail. The full name of the CP series has been explicitly defined at its first occurrence in the Simple Summary section. Additionally, both the full name and abbreviation are now clearly presented in line 56 of the Introduction.

Q2: The Reviewer is not sure if simple summary is required together with abstract. The question is for the Editor.

A: We thank the reviewer for raising this procedural matter. We confirm that the journal's guidelines specifically require submission of a separate Simple Summary document in addition to the main abstract.

Q3:PAM-2500 28 and SPAD instruments over three years. Our findings revealed significant variations in 29. It’s better to name the measured physiological parameters not the equipment used since the parameters not equipment are essential for the conclusions done.

A: We have carefully addressed this suggestion. The relevant content in Section 2.2 now explicitly lists the measured physiological parameters rather than the equipment used, as these parameters are indeed critical to our conclusions.

Q4: tion, utilizing native wild relatives (Saccharum spontaneum and Erianthus arundinaceus) 52 Botanical names to be in italics.

A: We appreciate this important editorial suggestion. The botanical names in the Introduction section have been properly italicized as per botanical nomenclature standards.

Q5: Hegde et al. demonstrated that leaf water deficit 79 in sugarcane leads to significant reductions in both variable fluorescence (Fv) and the 80 maximum photochemical efficiency of PSII (Fv/Fm_Ln). Reference to be added.

A: We thank the reviewer for identifying this omission. The reference to Hegde et al. has been added to the reference list and appropriately cited in the text.

Q6: The cane tops and 143 root systems were removed, retaining only the stalks for further analysis., It’s reasonable to indicate the lengths of the parts in cm or inches.

A: We appreciate this suggestion for greater precision. However, due to natural variations in growth among individual sugarcane stalks used in the experiment, exact measurements of the removed portions could not be consistently specified. We have therefore revised the description in Section 2.3 to “removal of canopy parts above the +1 leaf and underground portions” for clarity.

Q7: Table 1 to follow figure 1, not the opposite.

A: We have implemented this suggested change. Table 1 has been repositioned to immediately follow Figure 1 in the manuscript.

Q8:Figure 1. It’s reasonable to indicate the parameters reflected at the axes. 

A: We thank the reviewer for this suggestion. After careful revised, we have determined that all relevant parameters are already clearly displayed in the top-right section of Figure 1. These parameters are presented as dimensionless ratios or relative values, which we believe maintains scientific clarity while adhering to standard graphical conventions.

Q9:Figure 3. Please, provide more information how the clustering was done.

A: We appreciate this request for greater methodological transparency. The clustering algorithm used has been explicitly described in the opening sentence of Section 3.3.

Q10:Please, provide in the text of the Manuscript where the sugarcane genotypes are available; the same as mentioned in the replies to Reviewer.

A: We have implemented this suggestion. The sources of the sugarcane genotypes have been clearly detailed in Section 2.1 of the manuscript.

Q11:Figure S2. Please, indicate what’s SS here.

A: Perhaps referring to the unmarked position of Figure 2 in the text description. We have now clearly indicated its location in the manuscript.

Q12: while near-infrared spectroscopy data 147 were simultaneously collected with the MATRIX-F system (Bruker Optik GmbH, Ger-148 many)[19]. 149

Please, add which parameters were assessed.

A: We appreciate this request for greater detail. The specific parameters assessed through near-infrared spectroscopy have been added to Section 2.3.

Q13: Quality of all the figures could be improved.

A: We thank the reviewer for this suggestion. We have enhanced the resolution and overall quality of all figures and tables in the manuscript.

Q14: The MS essentially improved, the Reviewer suggests minor revisions.

A: We greatly appreciate the reviewer's positive assessment and have incorporated all suggested minor revisions throughout the manuscript.

Assistant Editor:

Q:Besides, we have noticed that Reference 23 has an associated correction. Could you replace it with another citation, or could you confirm that it is still acceptable to retain it without compromising the scientific validity? [23]Baker, N.R.; Rosenqvist, E. Applications of chlorophyll fluorescence can improve crop production strategies: an examination of future possibilities. /J Exp Bot/ 2004, 55, 1607-1621.

A: We sincerely appreciate the editor’s attention to the textual ambiguities. In relation to reference [23], the observed modification constitutes a necessary reordering resulting from the integration of additional citations within the manuscript. We have confirmed that this citation retains its relevance and substantiates the scientific rigor of our research.

This manuscript is a resubmission of an earlier submission. The following is a list of the peer review reports and author responses from that submission.

Round 1

Reviewer 1 Report

Comments and Suggestions for Authors

Present study investigates sugarcane photosynthetic traits of different genoptypes from various origins.  it provides critical insights for both cultivation management and genetic enhancement strategies, systematically elucidating the complex tripartite relationship between genotype, ratoon age, and germplasm origin in determining photosynthetic efficiency. The study demonstrates that genotype exerts the most profound influence on photosynthetic characteristics, with quantitative analysis revealing variation in key parameters among different cultivars, particularly highlighting the exceptional performance of certain genotypes in chlorophyll fluorescence and relative chlorophyll content, which directly correlates with sucrose accumulation potential. study addressed significance in genoptypic variation and thus provides important information. However, there are number of queries need to be addressed.

-Abstract is too general, it must be with specific results and conclusion.

-In introduction, there are number of mistakes in english, further it needs more details of the background and objectives.

-Materials and methods lacks husbandry of growing, there must be details of growing media conditions and interculture.

Soil analysis, fertilizer application, sources, detailed procedure for data recording are missing.

-Results are interpreted well, however more detailed discussion  needed.

-Conclusion is too long, it should be in small para.

Comments on the Quality of English Language

There are number of mistakes in text, some words are in repetition, need to be deleted. Grammatical errors also found. A thorough revision for english improvement is required.

Reviewer 2 Report

Comments and Suggestions for Authors

Introduction and Methods

These two sections have been poorly written, lacking detailed information, literature, and justification for the work. It leaves much to be desired.

The methodology, as it stands, lacks details of the experiment for repeatability. For instance, in 2.2, the author mentions daylight hours without specifying the exact time period during which the measurement was taken.

Title: Genotypic and Environmental Determinants of Photosynthetic Performance

in Sugarcane Across Multiple Ratooning Cycles.

What about fructose? Did the photosynthetic efficiency correlate with fructose content as well as sucrose, or was it only correlated with sucrose? It would be interesting to see how stalk weight and fructose content are related.

Figure 3 is a bar graph, not a box plot, and looks suspicious to me based on the error bars and ranking provided, and I request a full, detailed analysis of this figure.

Table 3 is poorly formatted and presented. Kindly review.

Table 4 presents the authors' ranking of differences in photosynthetic characteristics among different experimental genotypes, categorised as High, Moderate, and Low. However, a simple computation of their own values shows no significant difference.

E.g. Fv/Fo 3.621±0.117A (HPE) 3.426±0.083B (MPE) 3.297±0.075C

If you compute 3.621 - 0.117 = 3.504; 3.426 + 0.083 = 3.503. How is 3.504 significantly different from 3.503?

This is totally unacceptable.

Comments on the Quality of English Language

Alot of grammatical errors. 

Reviewer 3 Report

Comments and Suggestions for Authors

The Manuscript provides sufficient and even excessive volume of novel results to be considered for publication but it needs essential revisions before further evaluation for Journal Biology.

The present Reviewer suggests series of amendments and reject-resubmit to offer more time to Authors or major revision for the basically good Manuscript. Alternatively, it’s reasonable to submit the Manuscript to a more agricultural journal where it fits more.

General points.

1. The general trend of the Manuscript is to agriculture and economics but not to biology.

a. For a biological Manuscript the latin name of the species to be given.

b. Essential information about the whole procedure of ratooning to be given. What is it for, how is it done, what are the consequences for the plants etc. All the procedures should be introduced in the introduction and then well described in methods.

2. The methods are to be described more.

3. Namely, going on with point 2. The weather conditions to be provided, the type of soil, the origin of genotypes, their specific features for ratooning to be given. Alternatively, the reproducibility of the results poses dubious questions:

“Field trials employed a completely randomized block design with 99 single-row plots (5m length, 1.4m row spacing) at 16 shoots per meter. All genotypes were 100 replicated three times. The experiment was conducted at Guangxi University's Fusui Field 101 Station, with planting initiated in February 2021 (NP), harvesting conducted in January 102 2022, and retaining the first-year ratoon (R1), and second-year ratoon data collected in 103 2023 (R2).”

4. Going on with 2). The leaves selected for measurements to be given together with illumination conditions in mmoles/(m^2*s)

a. Field measurements were conducted during the sugarcane elongation phase (6 106 months post-planting/harvesting) under consecutive clear-day conditions. Fluorescence 107 parameters were assessed using a PAM-2500 portable modulated chlorophyll fluorometer 108 (Walz, Germany) between 20:00 to 23:00[9], while chlorophyll content was quantified with 109 an SPAD-502 Plus instrument (Konica Minolta, Japan) during daylight hours. Each geno-110 type was evaluated across ten plants with three technical replicates. 111

5. Going on with 2). What are the samples, was it the whole plant or part of the plants etc. Without the knowledge, it’s difficult to make any conclusions.

In December 2021 and December 2022, six representative sugarcane samples were 113 collected from each material during the sugarcane maturation period. After removing im-114 purities, the weight of the sugarcane was measured using an electronic scale, and the sin-115 gle-stalk weight of each genotype was calculated. The sugarcane stalks were crushed us-116 ing the DM540-CPS system (Sugarcane Crushing System, IRBI, Brazil), while near-infra-117 red spectroscopy data were simultaneously collected with the MATRIX-F system (Bruker 118 Optik GmbH, Germany)[18].

Minor numerous points, not all are given.

1) The study evaluated 74 sugarcane genotypes, including 20 international accessions 96 (USA, India, Australia) and 54 domestic cultivars from seven Chinese provinces: Guangxi 97 (26), Fujian (10), Guangdong (6), Yunnan (4), Taiwan (5), Hainan (2), and Sichuan (1) (Sup-98 plementary Table S1).

The journal is an international one, so have to be rewritten. Not international and domestic accessions but 20 accessions from USA, India, Australia and 54 cultivars from seven Chinese provinces.

The political question of Taiwan the present Reviewer leaves to the editors since it is mentioned not clearly https://en.wikipedia.org/wiki/Taiwan

2) Simple summary. The Reviewer suggest to check the Journal if the sort of summary is considered. Alternatively, it could be united with the abstract.

3) The CP series cultivars introduced from the United 20 States demonstrate superior photosynthetic efficiency, thereby presenting substantial 21

Please, indicate what is CP series and how the US are involved then.

4)

 photosynthetic traits of 74 sugarcane varieties using PAM-2500 30 and SPAD instruments over three years.

Please, indicate which parameters were measured not the equipment used.

5) Table 1. Please, indicate whether here is a representative example given for the lines or all the lines were measured in the same way but the results are not given. If all the lines were measured, then the other forms of presenting the results are better to provide in the Manuscript.

6) with planting initiated in February 2021 (NP), harvesting conducted in January 102 2022, and retaining the first-year ratoon (R1), and second-year ratoon data collected in 103 2023 (R2).

Please, indicate when the ratooning was completed and how, se above also.

7) Figure 1. Please, provide the units of measurements. The give the full description of how the count numbers are obtained.

8) Table 2 is good to be combined with the figures for the parameters measured and the groups for them.

9) Figure 2. Please, indicate how the groups were selected for the photosynthetic performance and provide the corresponding data.

10) Table 3. Please, provide the units of measurements.

11) Figure 2. Please, indicate the rationale behind the dendrogram, namely, how was it produced.

12) Figure 3. Please, indicate what the numbers below the main values mean. Please, provide the errors of the measured parameters.

13) Figure 3. Please, give all the abbreviations in full for axes of the figure.

14) Figure 3. Are here the error bars? Pls, provide the full statistical treatment.

15) The Reviewer was not able to reveal the original data for the genotypes.

16) The Reviewer would like to know the availability of the genotypes to reproduce the results if required. Are they available from the Authors?

17) Additionally, we found that ratoon age significantly affected the 233 photosynthetic traits of sugarcane.

The Reviewer would like to see more detailed description and figures for the statement.

18) Table S1. Information on the parentages of 74 sugarcane genotypes

What is the origin of all the genotypes then, how were they generated? The question about the availability of the genotypes had been asked earlier.

The Reviewer would like to see the much more detailed and extended text based on presumably the present results. The Reviewer suggests the Authors to start from the potentially huge volume of data present. There could be numerous models of photosynthetic efficiency based on the measured parameters. So far, the Authors produced very brief summary for the data which are not given and described in full. Therefore, it’s not possible for the Reviewer to make conclusions for the Manuscript.

Reviewer 4 Report

Comments and Suggestions for Authors

The manuscript "Genotypic and Environmental Determinants of Photosynthetic Performance in Sugarcane Across Multiple Ratooning Cycles" attempts to address photosynthetic traits of 74 sugarcane varieties and parameters of chlorophyll fluorescence and SPAD index over three years. In addition to some conceptual errors, such as stating that chlorophyll content was quantified with SPAD-502, as this equipment does not provide chlorophyll content, but rather a green color index, which is directly related to chlorophyll content, the manuscript's title suggests that environmental variables will be evaluated, but this is not the case. The research covered three years of sugarcane cycles, which were subject to different environmental conditions. The variables that make up chlorophyll fluorescence are influenced by the environment, so the results presented are meaningless if they are not related to the environment. Thus, the Conclusions are not based on robust Results, so I do not see a significant scientific contribution in this manuscript. The research fails to assess the genotype-environment interaction. In future manuscripts, avoid repeating words from the title in the Keywords.

Reviewer 5 Report

Comments and Suggestions for Authors

The manuscript addressed an important aspect of crop physiology, particularly photosynthetic activity, with direct relevance to improving productivity and sustainability of sugarcane production. The study is well-structured, and the experimental design appropriately supports the conclusions drawn. The authors need to revise the text to reduce the high similarity index in paragraphs.

The following points should be considered:

Lines 33-34: The authors mentioned the heritability of these traits ranged from 0.70 to 0.86, indicating a strong genetic influence, but didn’t put any table/figure in the results section.

Lines 47-93: The authors need to discuss physiological and morphological changes that occurred due to the effect of photosynthetic activities in sugarcane or other relevant crops. And need to discuss which environmental components severely affect photosynthetic activity with their genetic mechanisms.

Lines 113-114: The author mentioned that they collected data in December 2021 and 2022, but in the result section they demonstrated the results for 3 years of data (NP, R1 and R2).

Lines 149-151: Put the legend of the X-axis in the segmented graphs (A-G) in Figure 1.

Line 171: Create a PCA biplot main figure here and transfer this table to the supplementary figure.

Lines 186-188: These lines are the repetition of lines 176-178.

Lines 225-225: The authors need to make trait-wise correlation graphs, whether these trait(s) are involved in SSW and % of sucrose contents.

Lines 228-278: The authors need to explain the reasons for the obtained findings, highlighting the relevant previous findings.

Lines 279-298: In conclusion, the authors should mention their key findings.

The author needs to check the reference lists of those that are cited in the text. All the titles and other contents should be unique to the journal format.
